# The Role of Echocardiography in the Contemporary Diagnosis and Prognosis of Cardiac Sarcoidosis: A Comprehensive Review

**DOI:** 10.3390/life13081653

**Published:** 2023-07-29

**Authors:** Joseph Okafor, Rajdeep Khattar, Rakesh Sharma, Vasilis Kouranos

**Affiliations:** 1Department of Echocardiography, Royal Brompton Hospital, London SW3 6NP, UK; 2Cardiac Sarcoidosis Centre, Royal Brompton Hospital, London SW3 6NP, UK

**Keywords:** cardiac sarcoidosis, sarcoid, echocardiography, cardiac magnetic resonance, nuclear cardiology, FDG-PET, cardiac imaging

## Abstract

Cardiac sarcoidosis (CS) is a rare inflammatory disorder characterised by the presence of non-caseating granulomas within the myocardium. Contemporary studies have revealed that 25–30% of patients with systemic sarcoidosis have cardiac involvement, with detection rates increasing in the era of advanced cardiac imaging. The use of late gadolinium enhancement cardiac magnetic resonance and ^18^fluorodeoxy glucose positron emission tomography (FDG-PET) imaging has superseded endomyocardial biopsy for the diagnosis of CS. Echocardiography has historically been used as a screening tool with abnormalities triggering the need for advanced imaging, and as a tool to assess cardiac function. Regional wall thinning or aneurysm formation in a noncoronary distribution may indicate granuloma infiltration. Thinning of the basal septum in the setting of extracardiac sarcoidosis carries a high specificity for cardiac involvement. Abnormal myocardial echotexture and eccentric hypertrophy may be suggestive of active myocardial inflammation. The presence of right-ventricular involvement as indicated by free-wall aneurysms can mimic arrhythmogenic right-ventricular cardiomyopathy. More recently, the use of myocardial strain has increased the sensitivity of echocardiography in diagnosing cardiac involvement. Echocardiography is limited in prognostication, with impaired left-ventricular (LV) ejection fraction and LV dilatation being the only established independent predictors of mortality. More research is required to explore how advanced echocardiographic technologies can increase both the diagnostic sensitivity and prognostic ability of this modality in CS.

## 1. Introduction

Sarcoidosis is a multisystem chronic inflammatory disorder characterised by the presence of noncaseating granulomas within affected organs [1,2] The aetiology is unknown, but emerging evidence suggests that, in a person with genetic susceptibility, the disease is an immunological response to an antigenic trigger [2]. Cardiac involvement of sarcoidosis is considered rare, with clinically manifest cardiac involvement occurring in 5% of systemic sarcoidosis patients. However, autopsy studies have revealed that 25–30% of patients with systemic sarcoidosis have cardiac involvement, suggesting a greater burden of silent disease [3]. Treatment with corticosteroids and other immunomodulatory agents can slow down or avoid progression to heart failure, ventricular arrhythmias, and high-grade atrioventricular block [4].

The reported prevalence of cardiac sarcoidosis (CS) is increasing due to a notable step-up in new diagnoses following the incorporation of advanced cardiac imaging into routine clinical assessment [5]. Late gadolinium enhancement cardiac magnetic resonance (LGE-CMR) imaging and cardiac ^18^F-flurodeoxyglucose positron emission tomography (FDG-PET) have largely superseded the use of endomyocardial biopsy (EMB) in the diagnosis of CS [6]. EMB specimens demonstrating noncaseating epithelioid granuloma definitively establish a diagnosis [7]. However, due to the patchy distribution of granuloma within the myocardium, this technique confers a low sensitivity, and the procedure carries not insignificant risk outside expert hands [8]. In contrast, LGE-CMR imaging provides a detailed assessment of myocardial morphology including fibrotic changes in the subendocardial, mid-wall and subepicardial layers, whilst FDG-PET imaging can delineate areas of active myocardial inflammation.

Transthoracic echocardiography (TTE) is the most widely available and established non-invasive imaging modalities. TTE can be used throughout the patients’ diagnostic and management pathway, while the ability to identify CS patients at high risk due to LV impairment adds prognostic value. TTE has historically been used as a screening tool to trigger the need for advanced imaging when regional or global myocardial abnormalities are discovered. Over the last two decades, technological advancements including harmonic imaging and contrast echocardiography have driven improvements in image quality. Three-dimensional (3D) echocardiographic imaging now provides more accurate measurement of left-ventricular (LV) cavity volumes and ejection fraction (EF) [9]. The introduction of speckle-tracking strain analysis enables the earlier detection of regional and global subclinical myocardial dysfunction not only of the left ventricle, but also the left atrium (LA) and right ventricle (RV). Assessment of 3D LV strain now affords measurement of more complex myocardial mechanics including quantification of LV torsion and twist [10]. Such techniques have increased the diagnostic yield of echocardiography and provided important prognostic information.

The aim of this article is to provide a detailed contemporary review of the literature regarding the detection, surveillance, and prognostication of CS using conventional and advanced echocardiographic techniques.

## 2. The Current Role of TTE in Screening for Cardiac Sarcoidosis

The diagnosis of CS remains challenging due to the variation in disease severity and presenting symptoms, along with potential phenotypic overlap with other infiltrative disorders, cardiomyopathies, and coronary artery disease. However, due to the high morbidity associated with CS, screening patients with known extracardiac sarcoidosis (ECS) is important. The 2014 Heart Rhythm Society (HRS) consensus statement recommends screening on the basis of cardiac symptoms (palpitations, chest pain, presyncope, and syncope), electrocardiogram (ECG), and TTE [11]. The presence of either cardiac symptoms an abnormal ECG, or myocardial abnormalities on TTE warrants further investigation with LGE-CMR and FDG-PET imaging (Table 1). This approach carries a sensitivity and specificity of only 64% and 65%, respectively, for a CS diagnosis [12]. TTE as the sole screening test carries a 25–32% sensitivity and 95–100% specificity with 75% positive predictive value and 73% negative predictive value [13,14]. The positive predictive value of an abnormal TTE increases to 92% in a patient with ECS and cardiac symptoms [15]. However, due to its low overall sensitivity, in the presence of cardiac symptoms, LGE-CMR and FDG-PET imaging should be performed despite a negative TTE study. A combination of LGE-CMR and FDG-PET is considered the optimal imaging approach for the diagnosis of CS.

## 3. The Position of Echocardiography within Current Diagnostic Criteria

Over the past three decades, various clinical criteria for the diagnosis of CS have been published including the 1993 Japanese Ministry of Health and Welfare (JMHW) guidelines which were refreshed in 2006, the World Association for Sarcoidosis and Other Granulomatous Disorders (WASOG) 2014 guidelines, and the Japanese Circulation Society 2016 guidelines [16,17,18]. However, the 2014 HRS consensus recommendations are widely applied in clinical practice. In the absence of a histological diagnosis from EMB, a “probable” clinical diagnosis is obtained with the combination of biopsy-proven ECS and one of the following: advanced AV block, ventricular tachycardia, LVEF < 40%, LGE on CMR, or positive FDG-PET uptake [11]. With respect to TTE, the only applicable criterion is LVEF < 40%. However, the majority of CS patients at the time of diagnosis have preserved LV systolic function, thus limiting the sensitivity of conventional TTE [19]. The consensus document recognises that regional wall motion abnormalities (RWMA), scarring, or aneurysm formation on TTE should prompt LGE-CMR ± FDG-PET imaging, but does not regard these features as sufficient for a diagnosis of CS. Therefore, the HRS recommendations position conventional TTE as a screening tool and follow a multimodality imaging approach for the diagnosis of CS.

## 4. Echocardiographic Abnormalities in the Diagnosis of Cardiac Sarcoidosis

Several TTE abnormalities increase the likelihood of a patient with systemic sarcoidosis having cardiac involvement and should trigger LGE-CMR ± FDG-PET to confirm the diagnosis (Figure 1, central illustration). These include RWMA particularly when associated with wall thinning in a non-coronary distribution, most commonly seen in the basal septal region. Other abnormalities may include increased myocardial echogenicity and RV free-wall aneurysm formation.

### 4.1. Regional Wall Motion Abnormalities

The presence of a RWMA is an important signal of cardiac involvement. At the time of diagnosis, 35–50% of patients will have RWMA on TTE [20]. The LV free wall (96%) and the interventricular septum (73%) are the myocardial regions most commonly involved [21]. While regional wall thinning may be indicative of fibrosis and scar, thickened regions relate to granulomatous infiltration or oedema and can mimic other causes of asymmetric left-ventricular hypertrophy such as hypertrophic cardiomyopathy [21]. Increased echogenicity, especially involving the interventricular septum (IVS), provides a bright, speckled appearance of the myocardium (Figure 2). When present, this may reflect myocardial infiltration or active inflammation [22]. Focal abnormalities of the IVS can help the clinician distinguish between CS and other nonischaemic cardiomyopathies. A comparison between CS and idiopathic dilated cardiomyopathy (DCM) patients revealed a significantly higher rate of IVS thinning (<7 mm) or thickening (>13 mm) among the CS cohort (73% vs. 17%, *p* < 0.001) [23]. Contrast-enhancement agents such as Sonovue^TM^ have been used with abdominal ultrasound to better identify hepatosplenic sarcoid lesions [24,25]. In echocardiography, contrast may be used to more readily identify RWMA in those with poor acoustic windows. However, there is currently no data to suggest that its use increases the overall sensitivity of TTE in identifying CS.

Thinning and akinesis of the basal septum are among the more characteristic features of CS, present in 2–4% of patients with a CS diagnosis [20,26]. One study involving 175 CS patients and 2130 control subjects defined basal septal thinning as basal IVS width ≤ 4 mm and/or basal IVS-to-widest IVS ratio ≤ 0.6. Applying this definition to an ECS population, the presence of basal septal thinning carries 99% specificity and 39% sensitivity for a CS diagnosis [27,28]. When identified on a screening TTE of a sarcoidosis patient, in the absence of plausible alternative causes, cardiac involvement appears very likely.

### 4.2. Right-Ventricular Abnormalities

In CS, granuloma infiltration most commonly affects the LV rather than RV myocardium. Autopsy studies report a prevalence of RV involvement of 36–39% [29,30]. Recent clinical studies have demonstrated RV FDG-PET uptake in 9–22% of cases [31,32]. RV free-wall involvement on TTE can be hard to identify. TTE features of CS can mimic those of arrhythmogenic right-ventricular cardiomyopathy (ARVC) with RV free-wall thinning and aneurysm formation. When the diagnosis is unclear, measurement of LVEF is useful for distinguishing between the two diseases, with LV impairment (LVEF < 50%) more common in CS patients (53% vs. 7%, *p* < 0.001) [33]. RV involvement can lead to global systolic dysfunction. However, discriminating between direct myocardial involvement and RV impairment due to pulmonary hypertension from advanced respiratory or left-sided heart disease can be difficult. In the absence of pulmonary hypertension or significant diastolic dysfunction, RV impairment on TTE yields an 82–99% specificity for cardiac involvement [22]. Sensitivity, however, is low (10–47%) suggesting that global RV impairment alone is not a reliable marker for diagnostic purposes.

### 4.3. Valvular Abnormalities

Valvular dysfunction in CS is rare and usually secondary to chamber dilatation or papillary muscle inflammation causing functional atrioventricular valvular regurgitation (Figure 3). The prevalence of moderate to severe mitral regurgitation (MR) is approximately 11%, with Carpentier Type I functional MR by far representing the most common mechanism (46.3%), followed by Types II (22.2%), IIIb (20.4%), and IIIa (11.1%) [34]. Either anterolateral or posterolateral papillary muscle (PM) FDG avidity can be seen in 68% of CS patients with MR. However, sole avidity of the posterolateral PM is rare (3%) compared to sole anterolateral PM uptake (24%). Tricuspid regurgitation (TR) in CS is usually secondary to RV failure and subsequent tricuspid annular dilatation [35]. Rarely, direct valvular granulomatous infiltration within the tricuspid valve leaflets and moderator band can lead to severe TR [36]. In summary, valvular abnormalities on TTE are a nonspecific feature of CS but may indicate active inflammation when present.

### 4.4. Pericardial Abnormalities

Pericardial abnormalities on TTE can be seen in 19% of sarcoidosis patients [37]. Pericarditis secondary to direct involvement may result in small volume pericardial effusion which is typically serosanguinous or straw-coloured in appearance. Large volume effusions leading to tamponade are rare and are associated with extensive pericardial involvement [38]. Lastly, constrictive pericarditis as a sequalae of recurrent pericardial effusions may require pericardiectomy [39]. Care must be taken to differentiate this from the restrictive cardiomyopathy sometimes seen in CS patients with advanced myocardial disease.

### 4.5. Left-Atrial Abnormalities

Increased LA volume index may represent advanced LV myocardial disease associated with diastolic dysfunction, or the development of atrial arrhythmias. Direct sarcoid involvement of the LA is rare, with necropsy studies reporting a 7% prevalence [29]. Atrial wall thickening on TTE has been reported in a patient presenting with atrial flutter in the context of ECS and severe LV impairment [40]. The posterior LA wall was thickened with maximal width 3 mm on TTE. Transoesophageal echocardiography (TOE) revealed uniform thickening of all atrial surfaces including the interatrial septum, with low echogenicity compared to surrounding structures. A trial of anticoagulation did not affect the appearances. Separately, diffuse thickening of the LA wall with a maximal width of 4 mm on TOE corresponded with LA FDG uptake in a CS patient presenting with LA re-entrant tachycardia [41]. Such findings suggest closer attention to imaging the LA in sarcoidosis patients presenting with atrial arrhythmias, which may increase the sensitivity of echocardiography in diagnosing CS.

### 4.6. Left-Ventricular Strain

Myocardial deformation imaging is of value in the early detection of LV dysfunction in CS patients [42] (Figure 4). The complex cardiac architecture consists of three layers: subendocardial, mid-wall, and subepicardial. Early impairment of LV global longitudinal strain (GLS) represents the disruption of longitudinally organised myofibrils which are predominantly located in the subendocardial layer [43]. Classically, CS has a preference for epicardial and mid-wall involvement, as seen on LGE-CMR imaging. Nevertheless, CS patients still display impaired GLS at an early stage. One hypothesis for this was proposed by Kansal et al. [44]. The authors studied 59 patients with a range of ischaemic and nonischaemic cardiomyopathies including CS. They found that, independent of the distribution of LGE within the myocardial layers, GLS was markedly reduced compared to controls. This suggests that the burden of functional decline likely exceeds the structural involvement as identified by fibrotic scar.

GLS is significantly more impaired in patients with systemic sarcoidosis compared to healthy controls [42,45,46,47,48]. Table 2 summarises the current knowledge regarding the use of GLS to identify sarcoidosis patients with cardiac involvement. The main knowledge gap is related to the lack of studies with a high proportion of patients with CS, although Di Stephano et al. diagnosed 83 patients with CS using the HRS criteria [45]. The optimal GLS cut-off value for a CS diagnosis varies from −16.3 to −18%, depending on the strain software used. Strain has been proposed as a more sensitive measure of subclinical myocardial dysfunction than ejection fraction.

A significant reduction in LV GLS, even in the presence of preserved LVEF (LVEF > 50%), is associated with diagnosis of CS [49,50] (Figure 5). Therefore, strain analysis provides value in the screening process of sarcoidosis patients, over and above ejection fraction, and should be routinely incorporated into the TTE protocol.

**Table 2 life-13-01653-t002:** List of studies using LV GLS to predict cardiac involvement in sarcoidosis.

Study	Number of Sarcoidosis Patients	Number of Controls	Inclusion Criteria	Number of New CS Diagnoses	Criteria for CS Diagnosis	GLS Values (Sarcoidosis V Controls) or Cut-Off for CS Diagnosis	Sensitivity, Specificity and AUC for CS at Cut-Off. (NR = Not Reported)	Strain Software
Joyce, 2014 [43]	100	100	Patients with or without biopsy-proven extracardiac sarcoidosis	6	JMHW	−17.3% ± 2.5% vs. −20.0% ± 1.6%	NR	EchoPac
Murtagh, 2016 [50]	31	31	Biopsy-proven extracardiac sarcoidosis and preserved LVEF referred for CMR and TTE	31	LGE +	Cut-off for CS diagnosis −17%	Sens: 94% Spec: 94% AUC: 0.94	EchoInsight
Chen, 2018 [48]	54	54	Biopsy-proven extracardiac sarcoidosis, cardiac symptoms, and ECG changes	3	JMHW ± HRS	−16.8 ± 5.0 vs. −20.1 ± 3.2	NR	GE Vivid
Kusunose, 2019 [46]	139	52	Biopsy-proven extracardiac sarcoidosis referred for evaluation of CS	38	JMHW	Cut-off for CS diagnosis −18% (basal longitudinal strain)	Sens: 89% Spec: 69% AUC: 0.86	GE Vivid
Di Stefano, 2020 [45]	122	97	Biopsy-proven extracardiac sarcoidosis referred for evaluation of CS	83	HRS	Cut-off for CS diagnosis −16.3%	Sens: 82% Spec: 81% AUC: 0.91	GE Vivid

AUC = area under curve; CS = cardiac sarcoidosis; HRS = Heart Rhythm Society; JMHW = Japanese Ministry for Health and Welfare; LGE = late gadolinium enhancement; LVEF = left-ventricular ejection fraction; LV GLS = left-ventricular global longitudinal strain; Sens = sensitivity; Spec = specificity; TTE = transthoracic echocardiogram.

Circumferential strain values are derived from fibre shortening along the circular perimeter of the short axis of the myocardium [51]. During systolic ejection, simultaneous shortening of the oblique fibres in the left- and right-handed helices provide a horizontal counterforce. Circumferential performance is highly sensitive to injury within mid-wall myocardial fibres [52]. Consequently, the presence of mid-wall fibrosis tends to be associated with impairment of global circumferential strain (GCS) in non-ischaemic cardiomyopathies [53].

Regional circumferential strain values are significantly more impaired in the mid-myocardial and epicardial layers, compared to the endocardial layer [44]. In theory, therefore, among a pure CS population where mid-wall and subepicardial fibrosis dominates, GCS could be useful in diagnosis. Indeed, GCS values are significantly lower in asymptomatic patients with ECS compared to normal controls [47]. Furthermore, impaired segmental circumferential strain values derived by speckle-tracking echocardiography (STE) are known to correlate with areas of myocardial damage on CMR (LGE positive vs. LGE negative: −14 ± 5% vs. −28 ± 7%, *p* < 0.0001) [54]. Further prospective studies adhering to robust diagnostic guidelines are required to better understand the value of GCS in the screening and diagnosis of CS patients.

### 4.7. Right-Ventricular Strain

Impairment of RV systolic function, as measured by M-mode TAPSE and tissue Doppler imaging (S’), is a late sign in CS, representing significant RV free-wall scar or pulmonary hypertension. RV strain shows promise in identifying cardiac involvement of sarcoidosis at an earlier stage, but data are limited. Studies have compared RV strain parameters in suspected and confirmed CS patients against healthy controls [45,47]. RV GLS is derived from an average of strain values from the RV free wall and the right side of the IVS. RV free-wall strain (RV FWS) is a more specific measure of RV performance as the septal strain component is excluded, thus avoiding LV interaction. RV GLS, RV FWS, and RV global strain rate are all significantly reduced in CS patients. An RV GLS value of −19.9% provides 88.1% sensitivity and 86.7% specificity (AUC 0.93), while an RV FWS of −21.4% provides 86.4% sensitivity and 80.6% specificity (AUC 0.91). Care must be taken to adjust for the contribution of advanced diastolic dysfunction and subsequent pulmonary hypertension before attributing RV strain impairment to direct myocardial infiltration. The association between RV strain and RV FDG uptake is yet to be investigated. Due to the single-centre, retrospective nature of the above studies, caution should be applied when using the above cut-off values. Prospective studies are required to also understand the sensitivity of RV strain before pulmonary hypertension and RV dysfunction ensues, as well as the mechanistic link between impaired RV strain and LV myocardial disease in the absence of direct RV involvement.

### 4.8. Left-Atrial Strain

LA wall deformation is assessed in three different phases: reservoir (atrial diastole), conduit (passive ventricular diastole), and contractile phase (atrial systole). Approximately 40% of atrial function is provided by the reservoir phase compared to 35% and 25% for the conduit and contractile phases, respectively [55]. Peak atrial longitudinal strain represents the magnitude of atrial deformation and is analogous to LA GLS. Compared to healthy controls, sarcoidosis patients without cardiac involvement have significantly lower LA GLS values (34.3% ± 3.6% vs. 39.1% ± 4.1%, *p* = 0.001) [56,57]. Therefore, reduction in LA strain alone is an insensitive marker for the diagnosis of CS and should not be used in isolation.

### 4.9. Three-Dimensional Echocardiography

Three-dimensional transthoracic echocardiography (3DE) affords a more accurate and reliable appraisal of complex cardiac structures. 2DE relies on geometric assumptions to measure chamber size and EF, and there is increased inter-operator variability in measurements when endocardial definition is suboptimal. 3DE is less reliant on plane position and makes fewer assumptions about chamber size. Direct volume measurements are more accurate and reliable. For instance, the RV is a particularly challenging structure to assess due to its geometric complexity. 3DE has proven superiority over 2DE in assessing RVEF and RV end-diastolic volume [9]. In addition, 3D speckle-tracking strain has demonstrated incremental value in the assessment of myocardial mechanics. Additional deformation parameters such as torsion or twist can be evaluated (Figure 6). The ability to track speckles in two directions simultaneously affords measurement of global area strain. Limitations that exist with 2D STE such as through-plane motion are eliminated [10]. Only one prior study has investigated the role of 3DE in diagnosing CS [58]. The 3D speckle-tracking strain measurements were compared between CS and DCM patients. The 3D global radial strain (GRS) was significantly lower in the CS patients; however, 3D GLS and GCS measurements were similar. A GRS cut-off value ≤ 21.1% was able to distinguish between the two cardiomyopathies with a sensitivity of 70% and specificity of 88% (AUC 0.79). CS patients also displayed a greater number of negative radial strain curves reflecting the heterogenic nature of cardiomyocyte damage characteristic of CS compared to the global dysfunction of a DCM. Clearly, there is much scope to explore the potential of 3DE in better predicting subclinical cardiac involvement in sarcoidosis patients, and further prospective studies are vital before this technique becomes incorporated into routine clinical practice.

## 5. The Role of Echocardiography in Patients with Confirmed CS

The cornerstone of management in CS surrounds the identification and treatment of active inflammation, left-ventricular systolic dysfunction (LVSD), and arrhythmias. The disease course can be unpredictable; therefore, regular surveillance of CS patients is paramount, more so in those presenting with heart failure or a substantial inflammatory burden. FDG-PET plays a central role in identifying patients that would benefit from tailoring of immunosuppressive therapy and repeat imaging 6–9 months after initial induction therapy is recommended [6]. Further imaging thereafter is indicated if relapse of treatment failure is suspected [59]. Serial LGE-CMR imaging is useful for up-to-date risk stratification based on LGE burden, or if TTE evaluation of cardiac function is inadequate. Compared to FDG-PET and LGE-CMR, TTE is radiation-free and suitable for patients with poor renal function but lacks the additional extracardiac information afforded by cross-sectional imaging. Due to easy access, low cost, and high reproducibility, serial TTE is favoured for monitoring of LVEF. Where discrepancies in serial LVEF measurements arise, contrast-enhanced TTE or CMR is recommended. Another advantage of TTE over CMR is the lack of device artefact limiting image interpretation, although this is less of a concern in experienced CMR centres with highly developed MRI scanning protocols. In patients with advanced heart failure requiring mechanical support such as left-ventricular assist devices (LVADs), TTE is paramount and may guide adjustment in flow rates, as well as exclude intracardiac thrombus. There is no consensus regarding the optimal duration of serial LV function assessment, but our centre favours an annual evaluation of LVEF, or sooner in the setting of de novo heart failure symptoms. Breathlessness can be due to advancement in respiratory disease, LVSD, diastolic dysfunction, pulmonary hypertension, or new valvular disease, and TTE is crucial in distinguishing these disease states. A significant decline in LVEF should act as a trigger for FDG-PET imaging to look for disease relapse. In rare cases, new severe valvular regurgitation may indicate papillary muscle inflammation and should also trigger FDG-PET. Future studies are required to identify if advanced TTE modalities such as strain can reliably identify patients with new active inflammation, thus reducing the cost of a CS surveillance programme by reducing the number of FDG-PET scans performed.

## 6. Prognostic Value of Echocardiography in Cardiac Sarcoidosis

The major echocardiographic determinant of long-term outcome in CS remains LVEF. The sequelae of active inflammation followed by chronic myocardial fibrosis result in wall stiffening and initially regional systolic dysfunction. Multiple regions of hypokinesis eventually contribute to global systolic decline. Scarring and subsequent adverse remodelling (LV dilatation) of the myocardium further contribute to dyssynchronous and ineffective LV contraction. Increased severity of LVSD and greater LV end-diastolic diameter (LVEDD) are key determinants of mortality in CS patients, even following corticosteroid therapy [23]. Every 10 mm increase in LVEDD on TTE is associated with a 26% increase in mortality. In the era of advanced heart failure therapy such as cardiac resynchronisation therapy (CRT), LVEF remains important. Patients with severe (LVEF ≤ 35%) or moderate (36–50%) LVSD at the time of CS diagnosis have a poorer prognosis than those with preserved LVEF > 50% [60]. Crucially, while those with LVEF ≤ 35% are more likely to see some LV recovery following immunosuppression, most remain in the severely impaired category [5,60]. Current guidelines support the implantation of an ICD for primary preventative purposes in those with LVEF ≤ 35% (Class I) or in those with LVEF 36–49% (Class IIb) despite optimal medical therapy [11]. Regular TTE surveillance of LV function is, therefore, crucial to aid decision making on timing of device therapy in order to prevent sudden cardiac death. There is a lack of data on the prognostic value of RWMA per se. However, the presence of basal septal thinning at the time of CS diagnosis is associated with death, VA and heart failure hospitalisation, independent of corticosteroid therapy or CRT use [27]. In sarcoidosis patients, basal septal thinning is associated with future LVSD, even when LV function is preserved at time of detection [26].

Two prior studies have shown that a GLS value worse than −17% may be a useful prognostic marker in a general systemic sarcoidosis population with preserved LV systolic function [42,50]. In those with confirmed CS, heart failure hospitalisation correlates with a GLS lower than −14% in a population of 83 patients of whom only 23 had LVEF > 50% [47]. However, the study had a high (43%) prevalence of pulmonary hypertension, and failure to adjust for this and other important clinical factors meant alterations in myocardial mechanics by other comorbidities could not be excluded. Impaired regional GLS values correlates with areas of abnormal perfusion and metabolism on FDG-PET and is independently associated with adverse cardiac events [61]. There is a lack of data on the prognostic role of GCS or GRS in CS.

In CS, the RV is prognostically important as patients with RV LGE on CMR [62] or FDG uptake [31] are at higher risk of VA and death. On TTE, RV dysfunction usually signals pulmonary hypertension and/or direct RV myocardial infiltration. Contrary to CMR-derived RVEF, conventional TTE markers of RV function such as TAPSE, S’, and FAC have yet to demonstrate an association with adverse outcomes [63]. This likely reflects the limitations in acquiring these parameters such as the imaging plane, angle of the images, or inability to detect regional abnormalities in the RV free wall and septum. RV FWS is a more sensitive marker of functional deterioration and has incremental value over the conventional parameters listed above. In a direct comparison, RV FWS showed the highest AUC (0.74) to predict adverse events when paired against S’ (0.60), TAPSE (0.54), and FAC (0.48) [63]. Patients with RV FWS > −16.8% have poorer outcomes. In addition, those with impaired biventricular strain (basal GLS > −15% and RV FWS > −20%) show 5-year event-free survival of 71% compared to 100% with normal LV and RV strain [45].

Pulmonary hypertension remains an important cause of mortality in sarcoidosis patients, independent of other organ involvement. Sarcoidosis-associated pulmonary hypertension (SAPH) confers a 10-fold increase in mortality in sarcoidosis patients and has a prevalence of 18.8% using TTE [64]. The impact and prevalence in a confirmed CS population is unreported. The sensitivity and specificity of diagnosing SAPH using echocardiography varies from 69% to 100% and from 68% to 98%, respectively [65]. Correlation between right-heart catheterisation and TTE appears to be robust when the PASP is between 50 and 100 mmHg on Doppler assessment. However, parenchymal lung disease may obscure TTE windows, and not all patients have a measurable tricuspid regurgitant trace. Nevertheless, the estimation of PASP remains an integral part of the TTE protocol for CS patients.

## 7. Future Applications

TTE has a useful clinical role in CS due to its widespread availability and low cost relative to other imaging modalities. The potential of 3DE in the diagnosis and prognosis of CS patients has yet to be fully explored, particularly 3D speckle-tracking strain. Furthermore, deep learning (a machine learning method) algorithms have been developed for use on TTE images to predict the presence of various cardiomyopathies with high accuracy. To date, one study has developed a deep learning algorithm to distinguish CS patients from healthy subjects [66]. The pretrained algorithm was noninferior to five cardiologists in distinguishing echocardiographic images of CS from those of normal subjects. The potential exists to develop a model that can rapidly incorporate 2D and multi-chamber strain imaging in order to predict adverse cardiac events in CS patients.

## 8. Conclusions

Echocardiography plays an important role in the screening, surveillance, and management of CS patients. This role continues to grow due to advancements in echocardiography technology and software. Incorporation of myocardial strain analysis adds to the value of the study and should be considered routine. Detailed attention should be paid by the echocardiographer and clinician to specific abnormalities that increase the likelihood of CS and act as a trigger for more advanced imaging. A multimodality approach remains paramount when making the diagnosis and determining which patients would benefit from risk prevention measures. Nevertheless, challenges lie ahead. Currently, research has been largely confined to case–control and observational studies with large variation in the criteria used to diagnose CS. More prospective trials are required to fully understand the role of advanced echocardiography in all stages of decision making.

## Figures and Tables

**Figure 1 life-13-01653-f001:**
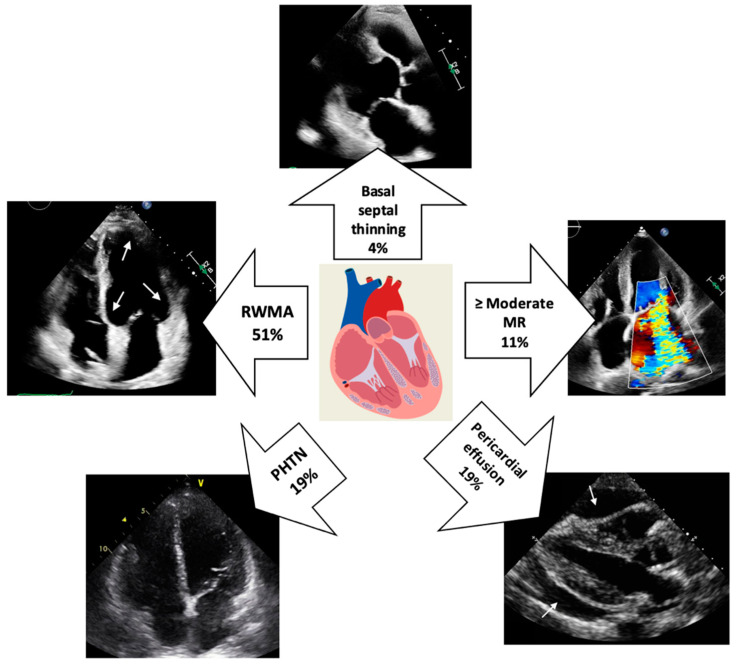
**Central illustration:** Prevalence of echocardiographic abnormalities in cardiac sarcoidosis. The white arrows indicate multifocal RWMA (top-left image) and global pericardial effusion (bottom-right image). (MR = mitral regurgitation; PHTN = pulmonary hypertension; RWMA = regional wall motion abnormality).

**Figure 2 life-13-01653-f002:**
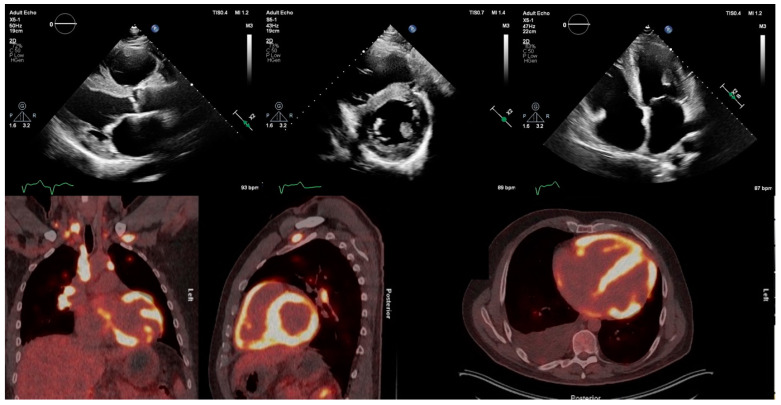
Transthoracic echocardiogram of patient with active cardiac sarcoidosis demonstrating hypertrophied, echogenic speckled left-ventricular myocardium in regions with high ^18^F-FDG uptake on PET, particularly the interventricular septum. ^18^F-FDG = ^18^F-fluorodeoxyglucose; PET = positron emission tomography.

**Figure 3 life-13-01653-f003:**
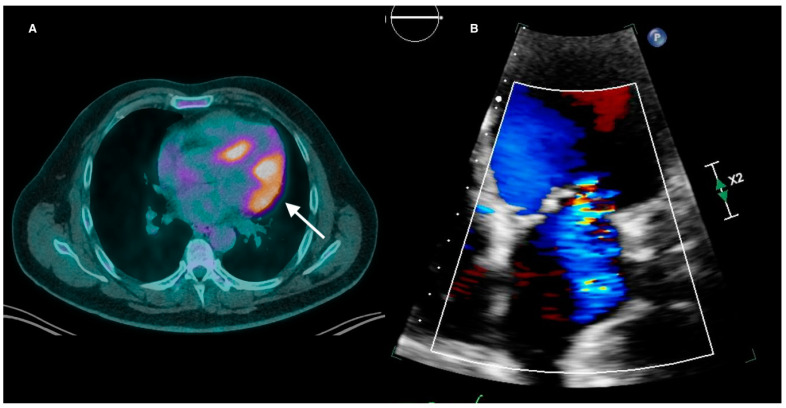
(**A**) Active cardiac sarcoidosis with FDG uptake extending from the lateral wall to involve the anterolateral papillary muscle (white arrow). (**B**) The same patient had moderate eccentric mitral regurgitation on transthoracic echocardiography. FDG = fluorodeoxyglucose.

**Figure 4 life-13-01653-f004:**
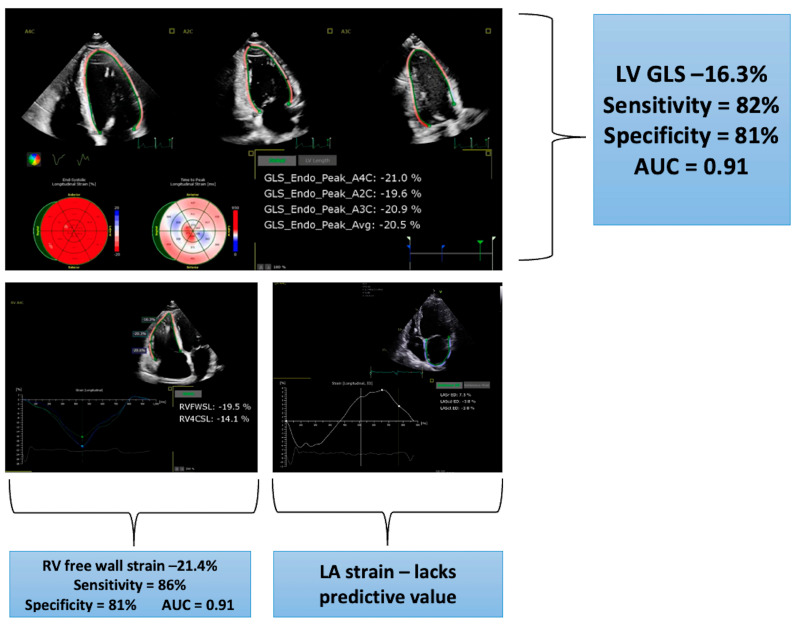
Impairment of multi-chamber myocardial strain to predict cardiac involvement in 122 sarcoidosis patients referred for cardiac evaluation. Optimal cut-off, sensitivities, specificities, and area under the curve (AUC) figures from a study into 83/219 sarcoidosis patients diagnosed with cardiac involvement [45]. LA = left atrial; LV = left-ventricular, GLS = global longitudinal strain, RV FWS = right-ventricular free-wall strain.

**Figure 5 life-13-01653-f005:**
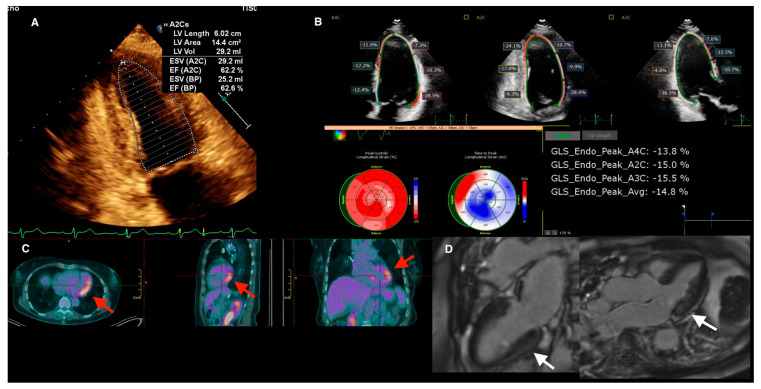
Multi-modality imaging in a 47-year-old patient with active cardiac sarcoidosis demonstrating the value of myocardial strain analysis, incremental to LVEF. (**A**) TTE revealed preserved LV systolic function with LVEF of 62%. (**B**) However, average GLS was impaired at −14.8% with segmental strain readings showing significant impairment in the lateral, basal inferior, and mid-apical anterior regions. (**C**) Areas of reduced strain correlate with regions of increased FDG uptake on PET (red arrows) and the CMR (**D**) revealed subepicardial LGE in the basal-mid inferior wall and mid-wall LGE throughout most of the lateral wall (white arrows). CMR = cardiac magnetic resonance; FDG = fluorodeoxyglucose; GLS = global longitudinal strain; LGE = late gadolinium enhancement, LVEF = left-ventricular ejection fraction; PET = positron emission tomography.

**Figure 6 life-13-01653-f006:**
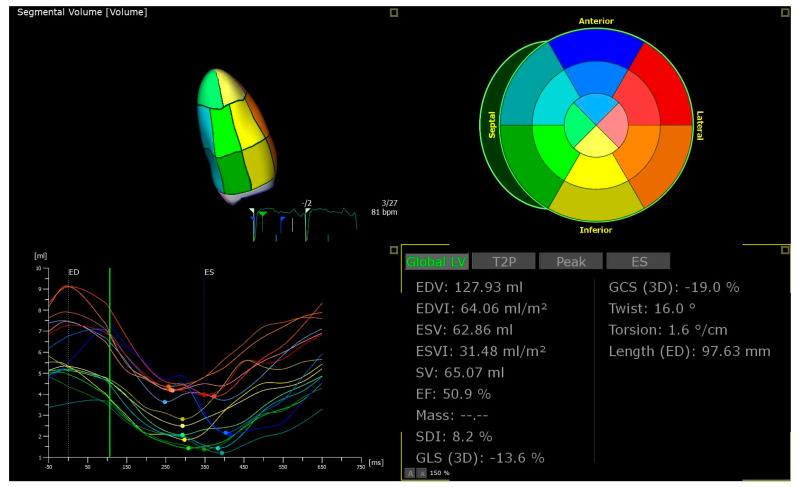
Three-dimensional speckle tracking echocardiography offers additional deformation parameters such as twist and torsion. ED = end-diastolic; EDV = end-diastolic volume; EDVI = indexed end-diastolic volume; EF = ejection fraction; ESV = end-systolic volume; ESVI = indexed end-systolic volume; GCS = global circumferential strain; GLS = global longitudinal strain; SDI = strain delay index.

**Table 1 life-13-01653-t001:** Summary of the role of echocardiography in cardiac sarcoidosis.

**Screening**	Conventional TTE lacks overall sensitivity (25–35%) but has high PPV (92%) in setting of cardiac symptoms and ECS.
	Abnormal TTE or normal TTE with cardiac symptoms warrants LGE-CMR ± FDG-PET investigation.
**Diagnosis**	TTE abnormalities include RWMA in noncoronary distribution (35–50%), RV impairment (19%), pericardial abnormalities (19%), ≥moderate MR (11%), and basal septal thinning (4%).
	Heart Rhythm Society guidelines incorporate unexplained LVEF < 40% + biopsy proven ECS as criteria for clinical CS diagnosis.
	Japanese Circulation Society guidelines incorporate septal thinning, ventricular wall aneurysm, or regional wall thickening as a major criterion.
	LV GLS, RV GLS, and RV FWS are significantly more impaired in sarcoidosis patients with cardiac involvement compared to those without.
	There is limited current research on the incremental diagnostic value of LA strain and 3D echocardiography.
**Surveillance**	TTE is crucial for serial monitoring of LVEF, diastolic function, valvular pathology, and emergent pulmonary hypertension.
	New deterioration in LVEF warrants further investigation for possible inflammatory recurrence.
**Prognosis**	Reduced LVEF and increased LV cavity dimensions are associated with adverse outcomes.
	Basal septal thinning presence on TTE is linked to increased mortality, ventricular arrythmias, and future LVEF decline.

3D = three-dimensional; CS = cardiac sarcoidosis; ECS = extracardiac sarcoidosis; FDG-PET = ^18^F-flurodeoxyglucose positron emission tomography; LA = left atrial; LGE = late gadolinium enhancement cardiac magnetic resonance; LV = left ventricular; LVEF = left-ventricular ejection fraction; LV GLS = left-ventricular global longitudinal strain; PPV = positive predictive value; RV = right ventricular; RV FWS = right-ventricular free-wall strain; RV GLS = right-ventricular global longitudinal strain; TTE = transthoracic echocardiography.

## Data Availability

No new data was created in the composition of this review article.

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
