# Peer review of "The Role of Echocardiography in the Contemporary Diagnosis and Prognosis of Cardiac Sarcoidosis: A Comprehensive Review"

_life, 2023, doi:10.3390/life13081653_

Round 1

Reviewer 1 Report

Thank you for the possibility to review the manuscript titled: “The Role of Echocardiography in the Contemporary Diagnosis and Prognosis of Cardiac Sarcoidosis: A Comprehensive Review”. The topic is interesting, important for modern cardiology and the manuscript is well-written and easy to read. The only minor recommendation is to add some information about USG with contrast enhancement (sonovue). There is limited data about its use in cardiac imaging but there are publications about sarcoidosis, other organs and USG with contrast enhancement:

Tana C, Schiavone C, Ticinesi A, Ricci F, Giamberardino MA, Cipollone F, Silingardi M, Meschi T, Dietrich CF. Ultrasound imaging of abdominal sarcoidosis: State of the art. World J Clin Cases. 2019 Apr 6;7(7):809-818. doi: 10.12998/wjcc.v7.i7.809. PMID: 31024952; PMCID: PMC6473121.

Tana C, Dietrich CF, Schiavone C. Hepatosplenic sarcoidosis: contrast-enhanced ultrasound findings and implications for clinical practice. Biomed Res Int. 2014;2014:926203. doi: 10.1155/2014/926203. Epub 2014 Aug 18. PMID: 25215299; PMCID: PMC4151864.

Please take into account the recommendations in the spirit of improving the quality of the submission.

Author Response

Thank you for the possibility to review the manuscript titled: “The Role of Echocardiography in the Contemporary Diagnosis and Prognosis of Cardiac Sarcoidosis: A Comprehensive Review”. The topic is interesting, important for modern cardiology and the manuscript is well-written and easy to read. The only minor recommendation is to add some information about USG with contrast enhancement (sonovue). There is limited data about its use in cardiac imaging but there are publications about sarcoidosis, other organs and USG with contrast enhancement:

Tana C, Schiavone C, Ticinesi A, Ricci F, Giamberardino MA, Cipollone F, Silingardi M, Meschi T, Dietrich CF. Ultrasound imaging of abdominal sarcoidosis: State of the art. World J Clin Cases. 2019 Apr 6;7(7):809-818. doi: 10.12998/wjcc.v7.i7.809. PMID: 31024952; PMCID: PMC6473121.

Tana C, Dietrich CF, Schiavone C. Hepatosplenic sarcoidosis: contrast-enhanced ultrasound findings and implications for clinical practice. Biomed Res Int. 2014;2014:926203. doi: 10.1155/2014/926203. Epub 2014 Aug 18. PMID: 25215299; PMCID: PMC4151864.

Please take into account the recommendations in the spirit of improving the quality of the submission.

Many thanks for your comments.

Now included in the subsection of RWMA: Contrast-enhancement agents such as Sonovue have been used with abdominal ultrasound to identify hepatosplenic sarcoid lesions [23,24]. For cardiac imaging, contrast may be used to better identify RWMA in those with poor acoustic windows. However, there is currently no data to suggest that its use increases the overall sensitivity of TTE in identifying CS.

Suggest references above also added.

Reviewer 2 Report

Firstly, I thoroughly enjoyed reading the review. The content was informative, and the use of English language was excellent. The images were nice.

1) I would like to recommend that the authors consider adding a table summarizing the main points or results of the review.

2) I would highlight the information value of ultrasound on ICD decision and SCD prevention.

Here are some specific notations:

Page 2: "18Fflurodeoxyglucose positron emission tomography 45 (FDG-PET)" should be corrected to "18F-fluorodeoxyglucose" with only "18" in superscript.

Figure 1: Please add 18F-FDG

Page 8: It appears that the font type/size changes between lines 260-1 (?)

Figure 5: Could there be an explanation provided for the abbreviations used in the figure?

Line 312: It would be beneficial to mention that the method is radiation-free (as 18F-FDG has a relatively high radiation dosage) and suitable for patients with poor renal function, unlike LGE. However, by using ultrasound you will lose information of possible extracardiac affisions.

Line 342: Please include references to support the statement that patients with LVEF ≤35% are less likely to experience left ventricular recovery following immunosuppression.

Author Response

Firstly, I thoroughly enjoyed reading the review. The content was informative, and the use of English language was excellent. The images were nice.

  • I would like to recommend that the authors consider adding a table summarizing the main points or results of the review.

Now included – please see table 1.

2) I would highlight the information value of ultrasound on ICD decision and SCD prevention.

Now reads: “Current guidelines support the implantation of an ICD for primary preventative purposes in those with LVEF ≤35% (Class I) or in those with LVEF 36-49% (Class IIb) despite optimal medical therapy [11]. Regular TTE surveillance of LV function is therefore crucial to aid decision making on timing of device therapy in order to prevent sudden cardiac death.”

Here are some specific notations:

Page 2: "18Fflurodeoxyglucose positron emission tomography 45 (FDG-PET)" should be corrected to "18F-fluorodeoxyglucose" with only "18" in superscript.

Actioned

Figure 1: Please add 18F-FDG

Actioned

Page 8: It appears that the font type/size changes between lines 260-1 (?)

After review, care has been taken to make sure the font is uniform across the whole manuscript

Figure 5: Could there be an explanation provided for the abbreviations used in the figure?

Actioned

Line 312: It would be beneficial to mention that the method is radiation-free (as 18F-FDG has a relatively high radiation dosage) and suitable for patients with poor renal function, unlike LGE. However, by using ultrasound you will lose information of possible extracardiac affisions.

Now reads: “Compared to FDG-PET and LGE-CMR respectively, TTE is radiation-free and suitable for patients with poor renal function but lacks the additional extracardiac information afforded by cross-sectional imaging.”

Line 342: Please include references to support the statement that patients with LVEF ≤35% are less likely to experience left ventricular recovery following immunosuppression.

Now reads: “Crucially, while those with LVEF ≤35% are more likely to see some LV recovery following immunosuppression, most remain in the severely impaired category [5,58].”

Reviewer 3 Report

Okafor et al. have conducted a comprehensive narrative review of the role of echocardiography in diagnosis and prognosis of cardiac sarcoidosis (CS). The authors reviewed traditional and newer echocardiographic techniques in the diagnosis and prognosis of CS. The manuscript is well-written, organized, and comprehensive. The authors highlighted echocardiography's role as a screening tool according to the HSR, JWHW, and WASOG guidelines.

The manuscript correctly points out limited utility of traditional echocardiographic findings such as regional wall motion abnormalities, global functional assessment (LVEF and RVEF), Left atrial abnormalities, and valvular regurgitation. Subsequently, the authors highlighted the utility of newer modalities, such as strain imaging and 3D echocardiography, in the diagnosis and prognosis of CS.

Major concerns:
1) Majority of the studies included in the review implemented a case-control design for a diagnostic test (echocardiography) evaluation. This limitation needs to be highlighted in the conclusion as well.  

The authors correctly identified the limitations of a case-control study design in the manuscript. For example, Line 247-249 "Further prospective studies adhering to robust diagnostic guidelines are required to better understand the value of GCS in the screening and diagnosis of CS patients." It needs to be reflected in the conclusion.

2) Conclusion is overstated. At present, echocardiography has a limited role in the diagnosis of CS, and prognostic parameters are not specific to CS. However, good study designs and newer echocardiographic modalities could improve the role of echocardiography in diagnosing CS.

Minor concerns:
1) It would be easier for the reader to follow the manuscript if authors could designate the usefulness (or lack of) of an echocardiographic modality at the end of each subheading, where ever applicable. For example, "Sensitivity, however, is low (10-47%) suggesting that global RV impairment alone is not a reliable marker for diagnostic purposes". Line 152.
2) Can authors report the sensitivity and specificity of GLS in Table 1? Studies with a case-control design may not have reported the sensitivity and specificity of GLS.
3) Line 245- Define (A full form) STE
4) Lines 261 and 262- Were higher sensitivity and specificity of RVGLS and RVFWS derived from references 43 and 44? If yes, those studies implemented a case-control study design in diagnostic studies, and the limitations of such design need to be highlighted.
5) Similarly, line numbers 294 and 295.
6) Line 305- Define LVSD
7) Line 320- "center."
8) First paragraph of the prognostic value of echocardiography in cardiac sarcoidosis- can authors clarify prognostic value of LVEF and LVEDD is not specific to CS?     
9) Determination of a "gold standard" or "reference test" in the evaluation of a diagnostic test in CS has always been a challenge. The authors have included multiple studies with different references against which echocardiographic modalities have been evaluated. It would be necessary to mention this to the readers.

Author Response

Okafor et al. have conducted a comprehensive narrative review of the role of echocardiography in diagnosis and prognosis of cardiac sarcoidosis (CS). The authors reviewed traditional and newer echocardiographic techniques in the diagnosis and prognosis of CS. The manuscript is well-written, organized, and comprehensive. The authors highlighted echocardiography's role as a screening tool according to the HSR, JWHW, and WASOG guidelines. 

The manuscript correctly points out limited utility of traditional echocardiographic findings such as regional wall motion abnormalities, global functional assessment (LVEF and RVEF), Left atrial abnormalities, and valvular regurgitation. Subsequently, the authors highlighted the utility of newer modalities, such as strain imaging and 3D echocardiography, in the diagnosis and prognosis of CS. 

Major concerns:
1) Majority of the studies included in the review implemented a case-control design for a diagnostic test (echocardiography) evaluation. This limitation needs to be highlighted in the conclusion as well.  

The authors correctly identified the limitations of a case-control study design in the manuscript. For example, Line 247-249 "Further prospective studies adhering to robust diagnostic guidelines are required to better understand the value of GCS in the screening and diagnosis of CS patients." It needs to be reflected in the conclusion. 

2) Conclusion is overstated. At present, echocardiography has a limited role in the diagnosis of CS, and prognostic parameters are not specific to CS. However, good study designs and newer echocardiographic modalities could improve the role of echocardiography in diagnosing CS.

Many thanks for your thorough consideration. The authors acknowledge these concerns and the conclusion now reflects this:

“Echocardiography plays an important role in the screening, surveillance and management of CS patients. This role continues to grow due to advancements in echocardiography technology and software. Incorporation of myocardial strain analysis adds to the value of the study and should be considered routine. Detailed attention should be paid by the echocardiographer and clinician to specific abnormalities that increase the likelihood of CS and act as a trigger for more advanced imaging. A multimodality approach remains paramount when making the diagnosis and determining which patients would benefit from risk prevention measures. Nevertheless, challenges lie ahead. Currently research has been largely confined to case-control and observational studies with large variation in the criteria used to diagnose CS. More prospective trials are required to fully understand the role of advanced echocardiography in all stages of decision making.”

Minor concerns: 
1) It would be easier for the reader to follow the manuscript if authors could designate the usefulness (or lack of) of an echocardiographic modality at the end of each subheading, wherever applicable. For example, "Sensitivity, however, is low (10-47%) suggesting that global RV impairment alone is not a reliable marker for diagnostic purposes". Line 152. 

This has been addressed and a mini synopsis added to the end of the RWMA, Valvular abnormalities, and 3DE subchapters.

2) Can authors report the sensitivity and specificity of GLS in Table 1? Studies with a case-control design may not have reported the sensitivity and specificity of GLS.

Added

3) Line 245- Define (A full form) STE

Actioned

4) Lines 261 and 262- Were higher sensitivity and specificity of RVGLS and RVFWS derived from references 43 and 44? If yes, those studies implemented a case-control study design in diagnostic studies, and the limitations of such design need to be highlighted. 

The end of the paragraph on RV strain now reads: “Due to the single-centre, retrospective nature of the above studies, caution should be applied when using the above cut-off values.”

5) Similarly, line numbers 294 and 295.

Now reads: “Clearly, there is much scope to explore the potential of 3DE in better predicting subclinical cardiac involvement in sarcoidosis patients and further prospective studies are vital before this technique becomes incorporated into routine clinical practice.”

6) Line 305- Define LVSD

Actioned

7) Line 320- "center."

Actioned

8) First paragraph of the prognostic value of echocardiography in cardiac sarcoidosis- can authors clarify prognostic value of LVEF and LVEDD is not specific to CS?     

While also common to other cardiomyopathies, LVEF and LVEDD were prognostic markers in a specific CS population. This has been made more explicit in the manuscript.

 Yazaki Y., Isobe M., Hiroe M., et al. Prognostic determinants of long-term survival in Japanese patients with cardiac sarcoidosis treated with prednisone. American Journal of Cardiology 2001;88(9):1006–10. Doi: https://doi.org/10.1016/S0002-9149(01)01978-6.)

9) Determination of a "gold standard" or "reference test" in the evaluation of a diagnostic test in CS has always been a challenge. The authors have included multiple studies with different references against which echocardiographic modalities have been evaluated. It would be necessary to mention this to the readers.

This has now been addressed in the conclusion (see response to point 2).